# Performance Improvement and Biofouling Mitigation in Osmotic Microbial Fuel Cells via In Situ Formation of Silver Nanoparticles on Forward Osmosis Membrane

**DOI:** 10.3390/membranes10060122

**Published:** 2020-06-16

**Authors:** Yuqin Lu, Jia Jia, Hengfeng Miao, Wenquan Ruan, Xinhua Wang

**Affiliations:** 1Jiangsu Key Laboratory of Anaerobic Biotechnology, School of Environmental and Civil Engineering, Jiangnan University, Wuxi 214122, China; luyiqin_zjjx@126.com (Y.L.); hfmiao@jiangnan.edu.cn (H.M.); wqruan@jiangnan.edu.cn (W.R.); 2Zhejiang Province Environmental Engineering Technology Appraisal Center, Hangzhou 310012, China; zhoukj@126.com

**Keywords:** osmotic microbial fuel cell, forward osmosis, membrane fouling, silver nanoparticle

## Abstract

An osmotic microbial fuel cell (OsMFC) using a forward osmosis (FO) membrane to replace the proton exchange membrane in a typical MFC achieves superior electricity production and better effluent water quality during municipal wastewater treatment. However, inevitable FO membrane fouling, especially biofouling, has a significantly adverse impact on water flux and thus hinders the stable operation of the OsMFC. Here, we proposed a method for biofouling mitigation of the FO membrane and further improvement in current generation of the OsMFC by applying a silver nanoparticle (AgNP) modified FO membrane. The characteristic tests revealed that the AgNP modified thin film composite (TFC) polyamide FO membrane showed advanced hydrophilicity, more negative zeta potential and better antibacterial property. The biofouling of the FO membrane in OsMFC was effectively alleviated by using the AgNP modified membrane. This phenomenon could be attributed to the changes of TFC–FO membrane properties and the antibacterial property of AgNPs on the membrane surface. An increased hydrophilicity and a more negative zeta potential of the modified membrane enhanced the repulsion between foulants and membrane surface. In addition, AgNPs directly disturbed the functions of microorganisms deposited on the membrane surface. Owing to the biofouling mitigation of the AgNP modified membrane, the water flux and electricity generation of OsMFC were correspondingly improved.

## 1. Introduction

The concept of an osmotic microbial fuel cell (OsMFC) was proposed in 2011 [1]. In an OsMFC, a forward osmosis (FO) membrane acts as a separator between the anode chamber with wastewater and the cathode chamber full of draw solution [1,2,3,4,5,6]. Specifically, the concentrated organic contaminants in the anode chamber by the FO membrane are oxidized with the production of bioelectricity, and water flux through the FO membrane transports protons from the anode to the cathode [1,2,3,4,5,6]. Compared with the conventional microbial fuel cell (MFC), the OsMFC achieves superior electricity production and better effluent water quality during municipal wastewater treatment [1,2,3,4,5,6]. However, there are some drawbacks hindering the application of OsMFC in wastewater treatment, including the lower water flux of the FO membrane, membrane fouling and salt accumulation [6,7]. Previous literature has demonstrated that membrane fouling has a significantly adverse impact on the water flux of the FO membrane and stable operation of the OsMFC [8,9,10]. In addition, membrane fouling reduces the electricity generation of MFCs owing to limiting the diffusion of cations [11,12]. Based on the critical influence of membrane fouling on the performance of the OsMFC, the fouling behavior of the FO membrane has been systematically investigated [5,6,8,9,10]. The results indicated that a thick fouling layer, including biofouling and inorganic fouling existed on the FO membrane surface in OsMFCs treating wastewater [8,9,10]. Lu et al. [7] further pointed out that biofouling was the dominant fouling type of the FO membrane in OsMFCs. Although these publications are helpful for understanding FO membrane fouling in OsMFCs, effective methods to mitigate fouling are still limited. Thus, it is necessary to search for effective fouling, especially biofouling, mitigation for improving the performance of the OsMFC.

According to previous studies [13,14,15,16,17,18,19,20,21], in situ modification of the membrane via biocides is an effective method to prevent the deposition of bacteria on the membrane surface and subsequently to mitigate the biofouling. Silver nanoparticles (AgNPs) are commonly used as the biocide owing to their strong antibacterial activity against numerous types of bacteria [13]. In recent years, several studies have demonstrated the potential of using AgNPs as the biocide to alleviate the biofouling in membrane separation processes including nanofiltration (NF) [14], reverse osmosis (RO) [15] and FO [16]. Generally, the protocol for Ag or AgNP deposition on membranes included embedding inside the membrane during membrane fabrication, and loading on the membrane surface via in situ formation [18,19,20,21]. For instance, Wu et al. [18] found that an Ag-PDA/PSf hybrid membrane achieved optimal separation performance because of dramatically enhanced protein-fouling resistance and good antibacterial activity, and Chew et al. [19] ascribed the membranes’ excellent performance to the coupled effects of improved surface hydrophilicity and a self-healing mechanism via in situ immobilization of the AgNPs. Based on the effective biofouling mitigation through modifying the membrane surface via AgNPs, it might be an alternative method to alleviate the biofouling of the FO membrane in OsMFC. In fact, the modified FO membrane made of cellulose triacetate (CTA) by AgNPs has been mentioned in OsMFC [17]. Given that the silver atom could not be directly formed on the pristine CTA–FO membrane due to lack of adhesion between AgNPs and substrate, the modified CTA–FO membrane was prepared by silver nitrate (AgNO_3_) with the help of dopamine and UV light. Although the AgNP modified CTA–FO membrane alleviated the biofouling, the aggravated concentration polarization of the FO membrane due to the addition of dopamine had an adverse impact on the performance of the OsMFC.

In order to avoid the diminishing of the CTA–FO membrane property modified by AgNPs, we intend to prepare the AgNP modified FO membrane using the thin film composite (TFC) polyamide FO membrane according to the following two reasons. Compared with the CTA–FO membrane, the TFC–FO membrane has more potential to be applied in wastewater treatment owing to its higher flux and selectivity, better pH stability, and resistance to hydrolysis [22,23,24,25]. In addition, AgNPs could be directly coated by sodium borohydride (NaBH_4_) on the TFC–FO membrane surface due to the presence of carboxyl groups on the membrane surface while AgNPs could not be directly formed on the CTA–FO membrane surface [13,26]. Thus, this study aims to mitigate the biofouling of the TFC–FO membrane via in situ formation of AgNPs on the membrane surface, and subsequently to improve the electricity generation of the OsMFC during wastewater treatment. To the best knowledge of the authors, this is the first study focusing on the modification of the TFC-FO membrane by direct coating with AgNPs for biofouling mitigation in OsMFC.

## 2. Materials and Methods 

### 2.1. In Situ Formation of AgNPs on TFC-FO Membrane

The dried TFC–FO membrane coupon (supplied by Hydration Technologies Innovations, Albany, GA, USA) was wetted through immersing in 20% isopropanol solution for 20 min. Subsequently, it was rinsed three times with deionized (DI) water and then placed in a design membrane module between a glass plate and a rubber frame for holding the solutions on the active layer of the TFC–FO membrane. The effective membrane area contacting the solution in the design module was about 80 cm^2^. In situ formation of AgNPs on the TFC–FO membrane was carried out in the following stages as summarized in Figure 1. Firstly, 20 mL of AgNO_3_ solution (2 mM) was put in the design module and directly reacted with the active layer for 10 min. Subsequently, the AgNO_3_ solution was discarded, and then the active layer of the TFC–FO membrane was reacted with a NaBH_4_ solution (2 mM) for 5 min. After forming AgNPs, the NaBH_4_ solution was removed from the membrane surface. Finally, the prepared AgNP modified TFC–FO membrane was rinsed for 10 s with DI water. All in situ reactions were done at ambient conditions of 25 ± 0.5 °C.

### 2.2. Analytical Methods

In order to identify the impact of in situ formation of AgNPs on the TFC–FO membrane properties, the pristine and modified TFC–FO membranes were characterized in terms of morphological observation, surface hydrophilicity, surface roughness, surface charge, anti-microbial properties and intrinsic separation characteristics. Specifically, a scanning electron microscope (SEM, S-4800, Hitachi, Minato-Ku, Japan) and an energy dispersive X-ray spectrometer (EDX, Falcon, EDAX Inc., Philadelphia, PA, USA) were applied for observing the membrane surface and identifying the AgNPs on the modified membrane surface, respectively. The surface hydrophilicity and the surface roughness were determined by a surface contact angle analyzer (OCA-15EC, Dataphysics, Stuttgart, Germany) and by an atomic force microscope (AFM, Bruker MuLtimode 8, Karlsruhe, Germany), respectively. In addition, the surface charge was analyzed by a SurPASS solid surface zeta potential analyzer (Anton Paar Co., Ltd., Glaz, Austria). The specific procedures for the SEM, EDX, contact angle analyzer, AFM and zeta potential analyzer can be found in previous literature [27,28]. The transport properties of the TFC–FO membranes were determined using a bench-scale FO system (8.5 cm × 3.9 cm) with DI water as the feed solution and 1 M NaCl as the draw solution [29,30]. The pure water permeability flux and the salt permeability coefficient were measured according to the method described in previous literature [28,31]. 

To quantify the AgNPs loaded onto the TFC–FO membrane surface, the modified membrane samples (1 cm × 1 cm) were firstly digested by 7% nitric acid at 105 °C for 2 h to promote dissolution of the silver from the membrane [32], and then the filtrate obtained by a 0.45 μm filter membrane was used for Ag analyses by an inductively coupled plasma mass spectrometer (ICP-MS, 720ES, Agilent, Santa Clara, CA, USA) [16]. In order to examine the residual silver loading on the membrane after the dissolution experiment, 4 cm^2^ coupons of in situ AgNP modified membranes were placed in 10 mL of 5 mM NaHCO_3_ solution (pH value of 8.3). After five days dissolution, the dissolved silver concentration in the solution was quantified with ICP-MS.

The antimicrobial property of the modified TFC–FO membrane was assessed using *E. coli* as the model bacteria [33,34,35]. An overnight-cultured bacteria in Luria Bertani (LB) medium was diluted to 10^8^ colony-forming units (CFU)/mL before using. A bacterial viability assay was carried out according to previous literature [36]. Specifically, the pristine and modified FO membranes were contacted with 1 mL bacteria solution, and then the membranes were gently rinsed and the remaining liquid was removed after 24 h incubation. A confocal laser scanning microscope (CLSM, LSM 710, ZEISS, Jena, Germany) was used for observing the distributions of dead and living cells on the pristine and AgNP modified membrane samples. In addition, the cytometry method was used to quantify the number of live bacteria attached to the FO membrane samples.

Water flux through the FO membrane was measured by the volume change of the draw solution versus time [8], and the reverse salt flux was calculated based on the conductivity change of the anolyte [37]. Silver ion concentrations of the anolyte were monitored at the end of experiment using ICP-MS (720ES, Agilent) after digesting by 7% nitric acid at 105 °C for 2 h, and the total organic carbon (TOC) was determined using a TOC analyzer (Shimadzu TOC-Vcsh, Kyoto, Japan). The measurements and calculations of TOC, ammonia nitrogen (NH_4_^+^–N), total nitrogen (TN), and total phosphorus (TP) concentrations in the FO permeate and their rejections by the FO membrane have been shown in our previous study [7]. In addition, the removal efficiencies of contaminants by the combination of microorganisms and FO membrane and only by microorganisms were also referred to in the previous study [7].

The foulants on the FO membrane surface were collected by ultrasonic (25 °C, 500 W, 20 kHz) for 30 min, and then their mixed liquor suspended solids (MLSS) and mixed liquor volatile suspended solids (MLVSS) concentrations were determined according to Chinese NEPA standard methods [38]. An EDX (Falcon, EDAX Inc., Philadelphia, PA, USA) and an SEM (S-4800, Hitachi, Minato-Ku, Japan) were applied for analyzing the element compositions and capturing the surface images of the fouled FO membranes, respectively. A CLSM (LSM 710, ZEISS, Jena, Germany) was used for observing the distributions of biofoulants, including microorganisms, proteins and polysaccharides on the fouled FO membrane samples. The specific methods of SEM, EDX and CLSM analyses have been reported in previous literature [17,39].

The OsMFCs voltages were recorded every 5 min by a data acquisition system (RBH8221, Ruibohua Co., Beijing, China). The polarization curve and internal resistance were obtained by using a series of different external resistances [40]. The volumetric densities of power and current were calculated based on the anode liquid volume [41].

All measurements were conducted at least three times, and the data were given with the mean value and the standard deviation.

### 2.3. Set-up and Operating Conditions

In order to compare the performance between the pristine and modified TFC-FO membranes, two identical laboratory-scale OsMFC set-ups (denoted as control OsMFC and AgNP-OsMFC, respectively) were operated in parallel. The schematic diagram of the set-up is shown in Appendix A. It consisted of an anode chamber and a cathode chamber (each with an effective volume of 144 mL), and the FO membrane was located between the two chambers. A carbon brush and a carbon cloth coated with Pt (0.3 mg/cm^2^) were pretreated as anode and cathode electrodes, respectively [7,40]. The pristine (supplied by Hydration Technologies Innovations, Albany, GA, USA) and the AgNP modified (made in house) TFC–FO membranes with an effective membrane area of 48 cm^2^ and an orientation of active layer facing the feed solution (AL-FS) were applied in the control OsMFC and the AgNP-OsMFC, respectively. The operation of both set-ups was stopped when their voltage was lower than 50 mV and then the solution in anode and cathode chambers would be replaced with fresh wastewater and 0.5 NaCl solution, respectively [7].

Both OsMFCs were operated under closed circuit conditions with 500 Ω external resistance at room temperature of 30 ± 0.5 °C during the whole experiment. The synthetic domestic wastewater was fed into the anode chamber, and its concentrations of TOC, NH_4_^+^–N, TN, and TP were 118.4 ± 4.0, 28.7 ± 0.5, 34.9 ± 1.5, and 2.91 ± 0.11 mg/L, respectively. The composition of the synthetic wastewater could be found in Appendix A. The 0.5 M NaCl solution was used as the draw solution. The anode and cathode chambers were circulated with a buffer tank at a cross-flow velocity of 0.03 cm/s. The seeded sludge in the anode chambers was collected from a local domestic wastewater treatment plant (Taihu Xincheng Wastewater Treatment Plant, Wuxi, China). The initial MLSS and MLVSS of the sludge in both OsMFCs were 3.2 and 2.4 g/L, respectively. 

## 3. Results and Discussion

### 3.1. In Situ AgNP Modified FO Membrane Analyses

The active layer images and EDX analyses of the pristine and modified TFC–FO membranes are illustrated in Figure 2. Compared with the pristine TFC–FO membrane, the surface color of the modified membrane changed from yellow (Figure 2(a-1)) to glossy dark brown (Figure 2(a-2)), and some particle clusters were observed on the surface of the modified membrane from the SEM image (Figure 2(b-2)). From EDX analyses (Figure 2(c-1),(c-2)), the pristine TFC–FO membrane only contained carbon, oxygen and sulfur elements, which is exactly consistent with the atomic components of the active layer, and silver (2.14%) was observed on the modified TFC–FO membrane surface (see Appendix A). The SEM and EDX results implied the successful formation of AgNPs on the modified membrane surface. Furthermore, the ICP-MS results indicated that the total quantity of silver on the modified TFC–FO membrane surface was up to 3.07 ± 0.10 μg/cm^2^. 

Characteristics of the pristine and modified TFC–FO membranes, including the surface contact angle, surface roughness, surface charge, water permeability (*J_w_*) and reverse salt flux (*J_s_*) were determined and compared for evaluating the impact of an AgNP coating on TFC–FO membrane properties (see Table 1). There was no significant change in the roughness between the pristine and modified TFC–FO membranes owing to the relative high roughness of the pristine TFC–FO membrane (Table 1 and Appendix A). We observed the smaller contact angle of the modified TFC–FO membrane (contact angle decreasing from 46.6 ± 1.3 degree to 33.6 ± 1.1 degree) (Table 1 and Appendix A). The current study suggested that the modified membrane had a better hydrophilicity [42,43,44,45]. As the hydrophilicity of the membrane surface increased, the adhesion of the pollutants would be poor [46,47]. The membrane surface charge was also considerably changed by the AgNP coating. The surface of the modified membrane (−34.56 ± 0.40 mV) had more negative charges than that of the pristine membrane (−19.10 ± 0.21 mV) at pH 7.5. This might be attributed to the fact that AgNPs have a negative zeta potential (approximately −20 mV) [48]. In general, the colloidal particles in water are negatively charged, and electrostatic repulsion between membrane surface and particles can reduce the adsorption and deposition of colloidal particles on the membrane surface. Therefore, more negative charges on the surface of FO membrane result in a better anti-fouling performance.

Additionally, it could be observed from Table 1 and Appendix A that water flux of the modified membrane was about 33.3% lower than that of the pristine membrane. Based on the fact that NaBH_4_ does not reduce the carboxylic or amides in the polyamide selective layer under the conditions used [49], the decrease in water permeability might be due to the deposition of AgNPs on the membrane surface resulting in a reduction of the effective membrane surface area for water flow [20,32]. However, *J_s_* was lower for the modified membrane (see Table 1), indicating that the nature of the polyamide layer was unaffected by the in situ reaction. Indeed, the *J_s_/J_w_* value of the modified membrane was much lower than the pristine membrane (see Table 1), suggesting that the modified membrane had better properties in the FO process.

In order to examine the effect of silver loading on the antibacterial activity, *E. coli* bacteria were contacted with the pristine and AgNP modified membranes for 24 h. A decrease of 25% in the number of *E. coli* bacteria colonies for the modified membrane compared to the pristine membrane was observed (see Table 1). Moreover, biofilm development was significantly suppressed on the modified membrane coupons indicated by the CLSM images (see Appendix A). These observations demonstrated that silver deposited on the TFC–FO membrane had a strong antibacterial ability.

The release of the AgNPs from the modified membrane to the aqueous solution was further evaluated. Based on the fact that silver ion dissolution from AgNPs highly depends on the solution chemistry and pH value, the modified membrane was immersed in a solution buffered by NaHCO_3_ at pH 7.5 (a similar pH value to the wastewater used in this study). After 5 days, only 18.99% of silver ion on the modified membrane was released, which proved that the AgNPs were strongly immobilized on the membrane surface.

### 3.2. Electricity Generation of OsMFC

The electricity generation of the control OsMFC and AgNP-OsMFC is illustrated in Figure 3. In both reactors, the voltage increased upon the replacement of the anolyte and then decreased due to the depletion of the organic substrates in all cycles. It is worth noting that the AgNP-OsMFC achieved a much longer operating time (approximately 760 h) than the control OsMFC (approximately 250 h), which could be attributed to a more severe flux decline of FO membrane in the control OsMFC. Moreover, the maximum voltage of AgNP-OsMFC was about 440 mV, which was higher than that in the control reactor (approximately 400 mV). In addition, the polarization test demonstrated the maximum power density of the AgNP-OsMFC and control OsMFC were 3.67 and 3.45 W/m^3^, respectively, and their internal resistances were approximately 305.8 and 303.7 Ω, calculated from the slope of the polarization curve, respectively. Based on the above results, the AgNP-OsMFC exhibited better electricity generation than the control OsMFC owing to better properties of the modified membrane. The improved hydrophilicity and highly negative charge of the modified membrane surface increased the proton or cationic ion transport thus reducing the internal resistance of the OsMFC [17]. Compared with the maximum power density of an OsMFC using the commercial CTA–FO membrane [7] and the modified CTA–FO membrane with Nag-pDA [17], a higher value was obtained in this study using the modified TFC–FO membrane.

### 3.3. Removal of Contaminants

Contaminants removal was also compared between the control OsMFC and the AgNP-OsMFC for evaluating the impact of the modified TFC–FO membrane on the performance of the OsMFC. Figure 4 shows the variations in TOC, NH_4_^+^–N, TN, and TP concentrations in the influent, anolyte, atholyte, and FO permeate during the operation of both OsMFCs. From Figure 4a, both reactors achieved an excellent TOC removal with a removal rate of more than 99.0%. It could be attributed to the combining effects of the microorganisms in the anode chamber (more than 96%) and the rejection of FO membrane (more than 99%). High TOC removal efficiency in this study was consistent with previous reports on OsMFCs treating low-strength wastewater [5,8]. Moreover, the average concentration of TP in the FO permeate in both OsMFCs was below 0.3 mg/L with a removal rate about 95% due to the high rejection of the FO membrane.

In addition, it could be seen from Figure 4b,c that the variations in NH_4_^+^–N of the influent, anolyte, atholyte, and FO permeate were consistent with that of TN because the main component of TN was NH_4_^+^–N. Thus, we only paid attention to the changes in NH_4_^+^–N. The FO permeate concentrations of NH_4_^+^–N in the control OsMFC and the AgNP-OsMFC were 20.7 ± 2.3 mg/L and 12.0 ± 1.4 mg/L with an average removal efficiency of 58.0% and 58.7%, respectively, which were similar results to a recent study on OsMFCs [50]. Like previous reports on OsMFCs [5,7,8], both OsMFCs in this study had better effluent quality compared to the traditional MFCs.

Since AgNPs might be released from the modified TFC–FO membrane to aqueous solutions, silver concentration in the anolyte was further assessed at the end of each cycle. The silver content in both anolyte and catholyte was below detection limit (<0.01 μg/mL), indicating that AgNP release was negligible and had little influence on microorganisms. 

### 3.4. Performance of the FO Membrane

Variations in the FO membrane flux and the conductivity of anolyte and catholyte at the end of each cycle of both OsMFCs are presented in Figure 5. It could be observed that the modified FO membrane had a larger initial water flux (4.55 LMH versus 3.49 LMH) and a lower flux decline rate (2.87 × 10^−3^ LMH/h versus 8.69 × 10^−3^ LMH/h) than the pristine FO membrane. It implied that the modified FO membrane achieved a better flux performance in the OsMFC. With regard to the variations in salinity in each cycle of both OsMFCs, it gradually increased in the anode chamber due to the rejection of the FO membrane for influent solutes and the reverse solute transport from the draw solution while it decreased in the cathode chamber owing to the dilution of the FO permeate. The variations in salinity in the anolyte and catholyte of both OsMFCs were consistent with previous literature [7]. It can also be seen from Figure 5 that there was no significant difference in the final salinity in both anolyte and catholyte at the end of each cycle between the two OsMFCs, suggesting that the modified FO membrane had no significant impact on the salinity variations in OsMFC. Considering that the variations in FO membrane flux in OsMFC were dependent on salinity and membrane fouling [7], the better flux performance of the modified FO membrane could be attributed to the mitigation of membrane fouling on the basis of a similar salinity (in the range of 15–30 mS/cm) in both reactors. 

### 3.5. Mechanisms of Fouling Mitigation by AgNPs

In order to understand the role of AgNPs in mitigating biofouling, the FO membranes were removed from the control OsMFC and AgNP-OsMFC. The photo and SEM images (see Figure 6) showed an apparently different morphology of the fouled TFC-FO membranes in the control OsMFC and in the AgNP-OsMFC. The fouled TFC–FO membrane in the control OsMFC was covered with a thick cake layer (with thickness of 69.87 ± 2.67 μm measured by the CLSM), while a thin fouling layer (with thickness of 58.19 ± 1.24 μm measured by the CLSM) was found on the modified TFC–FO membrane in the AgNP-OsMFC. Furthermore, the EDX results (see Figure 6) indicated a great variety of elements in the fouling layer of both TFC–FO membranes including C, O, Na, Mg, Al, P, S, Cl, K, Ca, Fe and Cu, which could be found in the feed wastewater. Thus, influent wastewater was regarded as the main source of the membrane foulants. The existence of metal cations such as Ca^2+^ and Mg^2+^ demonstrated the combined fouling on both FO membrane surfaces [51,52]. In order to distinguish the different contributions of inorganic fouling and biofouling, the foulants on the FO membrane were further collected and analyzed according to the method described in previous reports [53,54]. The MLVSS/MLSS rate of the foulants on the control and modified TFC–FO membranes were 0.90 ± 0.08 and 0.88 ± 0.07, respectively (see Appendix A), suggesting that biofouling played a more significant role in the fouling of TFC–FO membranes during the operation of the OsMFC. 

In order to further understand the biofouling of FO membranes in both OsMFCs, variations in α-d-glucopyranose and ß-d-glucopyranose polysaccharides, proteins, and microorganisms on the FO membrane surfaces were investigated by the CLSM combined with the multiple fluorescence probes, and their biovolume was calculated by the software of PHLIP. As shown in Figure 7 and Appendix A, the fouling layer of the modified TFC–FO membrane was thinner than the pristine membrane (58.19 ± 1.24 µm versus 69.87 ± 2.67). In addition, the biovolume of the polysaccharides, proteins, and microorganisms was less in the fouling layer of the modified TFC–FO membrane (Figure 7 and Appendix A), indicating that the organic foulants and biofoulants were reduced on the TFC–FO membrane via the modification by AgNPs. Furthermore, the mechanism of the AgNPs mitigating biofouling layer could be divided into two parts. On the one hand, the improved performance of the modified TFC–FO membrane was due to the increased hydrophilicity and the more negative zeta potential decreasing the fouling tendency of the FO membrane. In this case, the microorganisms and their secretion of extracellular polymeric substances were hard to deposit on or attach to the modified TFC–FO membrane surface as a result of the increased repulsion between foulants and the membrane surface [18,19,20,21]. On the other hand, with their antibacterial property, AgNPs disturbed the functions of the microorganisms deposited on the FO membrane surface and eventually led to the death of the microorganisms [18,19,20,21]. 

## 4. Conclusions

The AgNP modified TFC–FO membrane successfully mitigated biofouling in OsMFC. It could be attributed to the changes of TFC–FO membrane properties and the antibacterial property of AgNPs on the membrane surface. The modified TFC–FO membrane achieved an increased hydrophilicity and a more negative zeta potential which enhanced the repulsion between foulants and the membrane surface. In addition, in situ formation of AgNPs on the TFC–FO membrane surface could effectively disturb the functions of microorganisms deposited on the membrane surface. The biofouling mitigation of the FO membrane further improved the water flux and the electricity generation of the OsMFC.

## Figures and Tables

**Figure 1 membranes-10-00122-f001:**
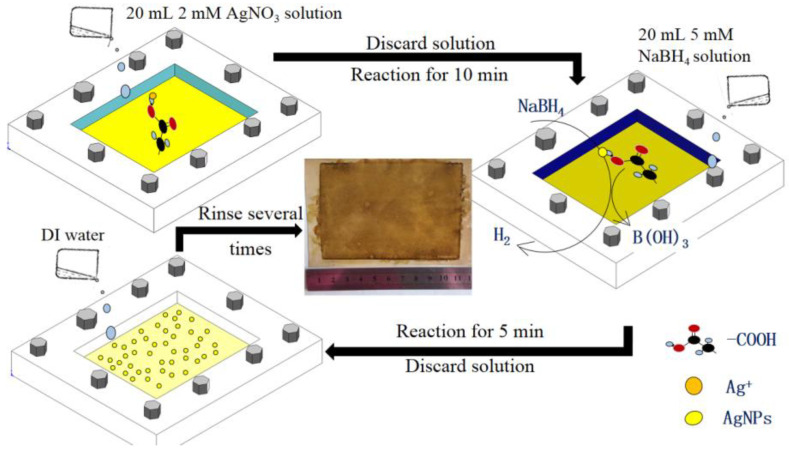
Sketch map of the preparation for an AgNP modified thin film composite-forward osmosis (TFC–FO) membrane.

**Figure 2 membranes-10-00122-f002:**
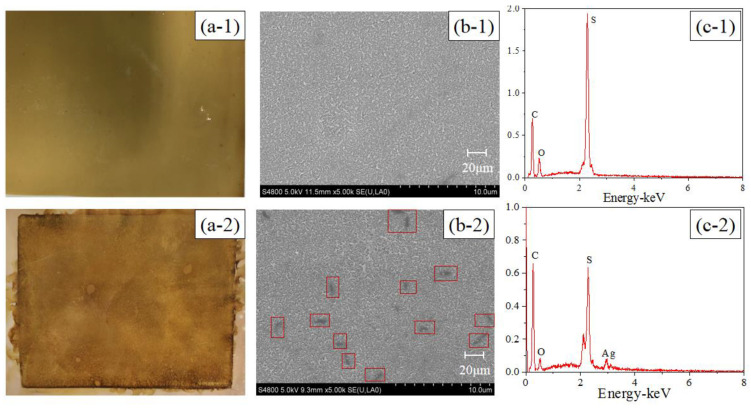
(**a**) Photographic images, (**b**) SEM images, and (**c**) EDX analyses of (1) the pristine and (2) the modified active layer of the TFC–FO membranes.

**Figure 3 membranes-10-00122-f003:**
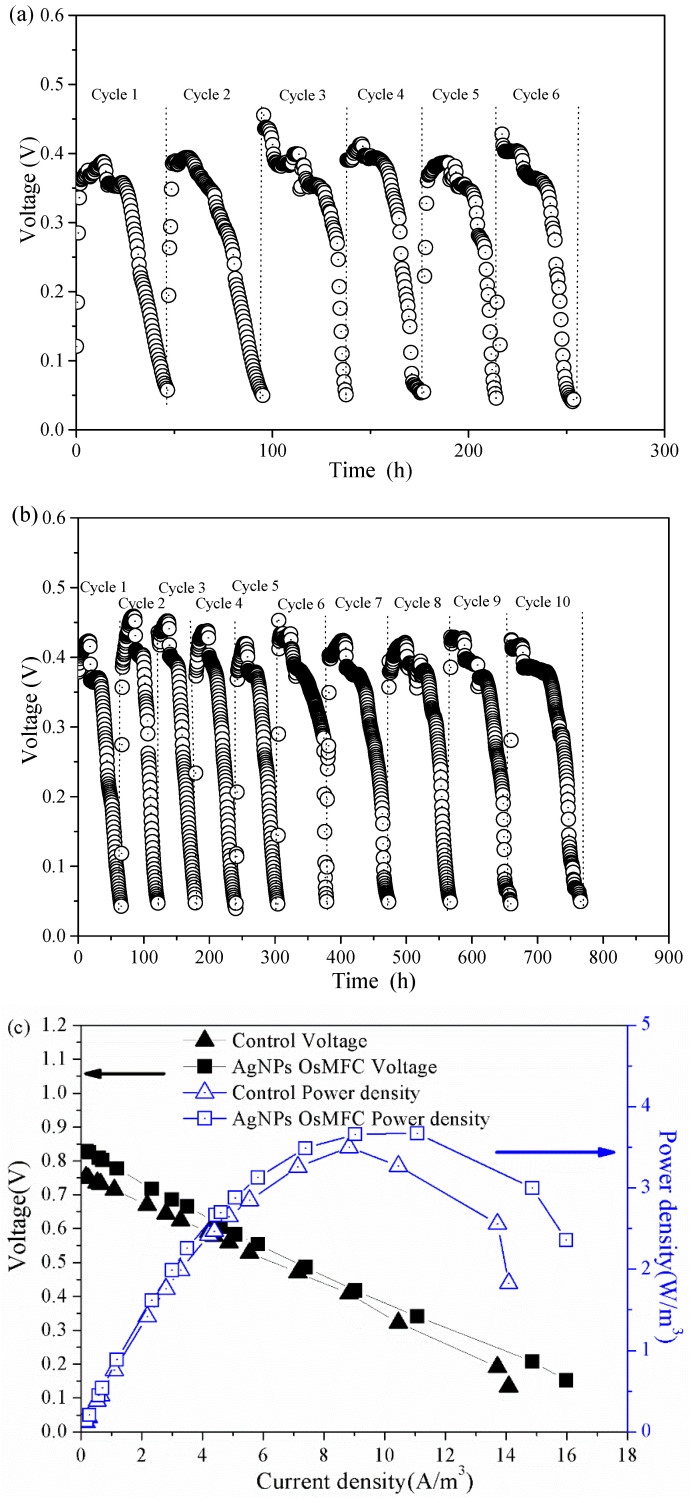
Electricity generation of (**a**) the control OsMFC and (**b**) AgNP-OsMFC, and (**c**) their power density and polarization curves.

**Figure 4 membranes-10-00122-f004:**
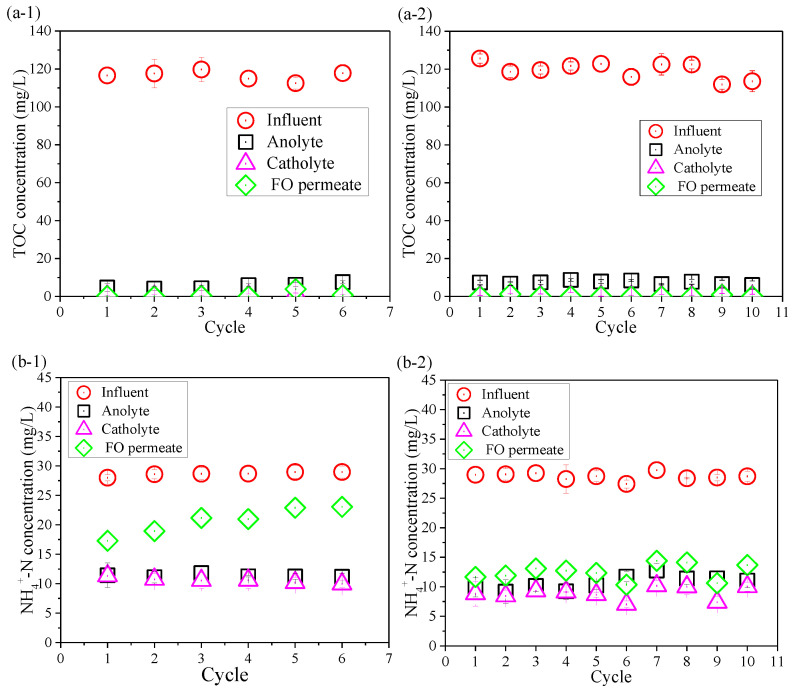
Variations in (**a**) total organic carbon (TOC), (**b**) NH_4_^+^–N, (**c**) total nitrogen (TN), and (**d**) total phosphorous (TP) concentrations in the influent, anolyte, catholyte, and FO permeate at the end of each cycle in (1) the control OsMFC and (2) the AgNP-OsMFC, respectively.

**Figure 5 membranes-10-00122-f005:**
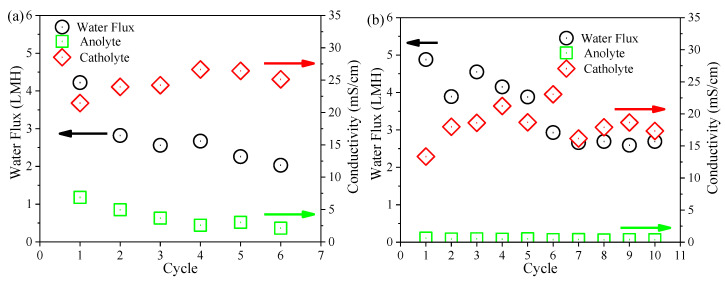
Variations in water flux of the FO membrane and conductivity of the anolyte and catholyte at the end of each cycle in (**a**) the control OsMFC and (**b**) AgNP-OsMFC.

**Figure 6 membranes-10-00122-f006:**
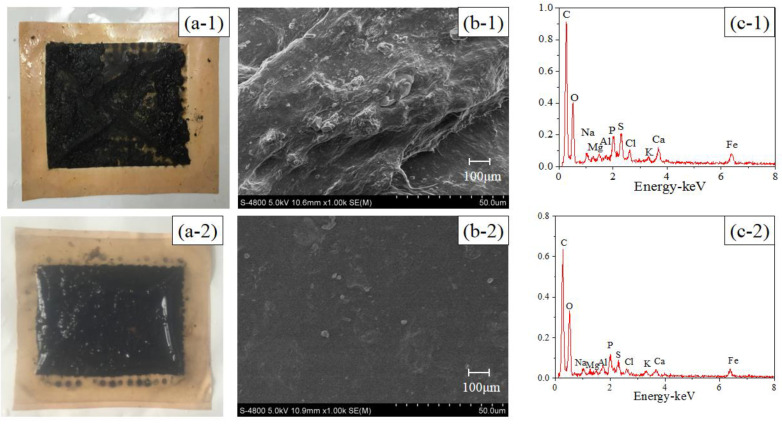
Photos (**a**) SEM and (**b**) EDX; (**c**) results of (1) the fouled pristine membrane and (2) the modified TFC–FO membrane.

**Figure 7 membranes-10-00122-f007:**
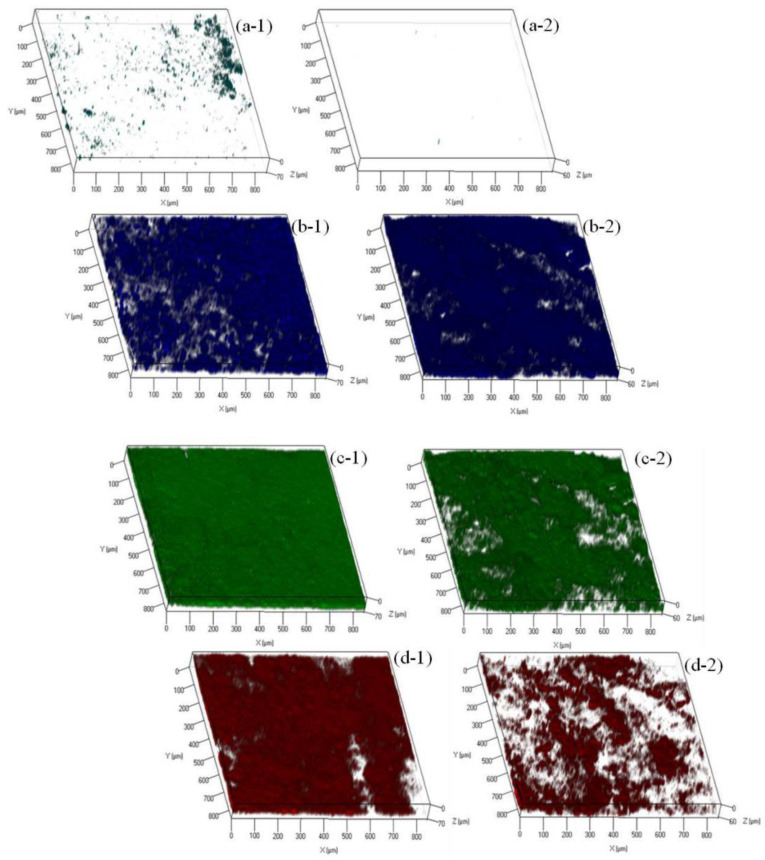
Confocal laser scanning microscopy (CLSM) images of (**a**) α-d-glucopyranose polysaccharides, (**b**) ß-d-glucopyranose polysaccharides, (**c**) proteins, and (**d**) microorganisms in the control (**1**) OsMFC and (**2**) AgNP-OsMFC.

**Table 1 membranes-10-00122-t001:** Characteristics of the pristine and modified TFC–FO membranes ^a^.

Characteristics	Pristine Membrane	Modified Membrane
Contact angle (degree)	46.6 ± 1.3	33.6 ± 1.1
Surface charge (mV)	−19.10 ± 0.21	−34.56 ± 0.40
Surface roughness (nm)	38.4 ± 1	33.3 ± 2
Water flux (LMH)	12.5 ± 0.5	8.5 ± 0.5
Reverse salt flux (10^−8^ m/s)	9.4 ± 0.3	2.7± 0.1
*J_S_*/*J_W_*	0.75 ± 0.12	0.31 ± 0.05
Bacteria colonies (10^7^ CFU/mL)	2.05 ± 0.07	1.55 ± 0.07

^a^ Values are given as mean values ± standard deviation (number of measurements: *n* = 3).

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
