# Peer review of "Performance Improvement and Biofouling Mitigation in Osmotic Microbial Fuel Cells via In Situ Formation of Silver Nanoparticles on Forward Osmosis Membrane"

_membranes, 2020, doi:10.3390/membranes10060122_

Round 1

Reviewer 1 Report

The manuscript is discussing the impact of silver nanoparticle deposition on FO membrane for OsmFC applications and with a special objective to mitigate biofouling. The whole study is of interest and covers both membrane synthesis/characterization and tests in application and making a clear baseline/modified membrane comparison. Still I have some important concerns regarding this study especially with regards to the reproducibility/replicates done since everything seems to be based just on one test. Before being consider for publication the authors should address this point, explicate methodology for each step, number of replicates and standard deviation. Otherwise the value of this study remains limited.

Apart from this major concern I have the following comments:

  • English is average, revision by a native speaker would help the understanding.
  • Several double blanks through the document, please revise
  • L56: ref 15 title refers to copper nanoparticles, not silver.
  • L70: interest of TFC vs. CTA is more connected to higher flux /tunability and selectivity
  • L66-77: more extended review on paper/protocol for Ag deposition on membranes would be welcome. Also is the method you applied taken from the literature?
  • L80: description missing. commercial or homemade TFC membrane? if home-made explain procedure or ref article, if commercial tell type/company…
  • L111: standard method? ref?
  • Mat & method: how many sample/ replicate for each membrane synthesis/ analyses/ each tests. This should be explained extensively and you should provide standard deviation in the result section
  • L127: schematic diagram, please provide at least in supplementary
  • L139: WW composition, please provide at least in supplementary
  • Figure 2: how many tests/images/surf analyses were done? also are the patches supposed to be silver particles? then it may tell that surface deposition is far from perfect.
  • L206: membrane surface area?? you mean permeability? there is an obvious loss of permeability and increase of selectivity. Please check in the literature (not only FO) if this is a typical effect and the reason for that. Pure permeability decrease due to the new layer deposition or due to interactions?
  • L211-214: Ecoli: please add some results/fig in the manuscript. repeats? how many samples? std deviation, etc...? results compare to literature?
  • L221-222: I do not understand that. what do you mean by longer operating time? how did you fix your stopping point/parameter?
  • L281: salinity may change as well since the modified membrane had higher selectivity to salt. You cannot really discriminate both effects just based on that test.
  • L290: did you try to quantify the amount of foulant? should appear in the manuscript
  • L298-299: statistically different? replicates?

Author Response

A point-by-point response to the reviewer’s comments has been uploaded as a Word file.

Reviewer 2 Report

This manuscript investigates the performance improvement and biofouling mitigation in osmotic microbial fuel cells via in situ formation of silver nanoparticles on a forward osmosis membrane. An array of characterization techniques were employed with convincing evidences provided to support the experimental observations and fouling-resistant property of the modified membrane. For these reasons, I recommend the publication of this manuscript with the following revisions:

  1. This manuscript generally reads fine but I strongly recommend that the authors further improve on its grammar and overall style. Please make the recommended changes as follows:

Line 38, change ‘applications’ to ‘application’.

Line 41, change ‘reduced’ to ‘reduces’

Line 45, change ‘was existed’ to ‘existed’.

Line 47, change ‘were helpful’ to ‘are helpful’.

Line 48, change ‘were still limited’ to ‘are still limited’.

Line 49, change ‘mitigations’ to ‘mitigation’.

Lines 51–53, please include citation(s).

Line 53, add silver nanoparticles in front of ‘AgNPs’.

Line 62, add silver nitrate in front of ‘AgNO3’.

Line 66, change ‘reduction’ to ‘diminishing’.

Line 69, change ‘is more potential to applying in’ to ‘has more potential to be applied in’.

Line 71, add sodium borohydride in front of ‘NaBH4’.

Line 76, change ‘mitigations’ to ‘mitigation’.

Lines, 82, 84, and 86, change ‘designed module’ to ‘design module’.

Line 86, ‘isolated active layer’ means the active layer is removed from the polysulfone support and polyester backing. Please use a better phrase to avoid confusion.

Line 116, change ‘An-overnight-cultured’ to ‘An overnight-cultured’.

Line 130, change ‘coasted’ to ‘coated’.

Line 191, please rephrase ‘lower than 27.9% values’.

Line 200, change ‘negative charge’ to ‘negative charges’.

Line 200, change ‘anti-pollution’ to ‘anti-fouling’.

Lines 240, 248, 263, 267, 271, 274, 279, 283, 309, and Figure S5, change ‘variations of’ to ‘variations in’.

Line 250, change ‘attentions’ to ‘attention’.

Line 250, change ‘changes of’ to ‘changes in’.

Line 253, change ‘similar as’ to ‘similar to’.

Line 253, change ‘likely’ to ‘like’.

Lines 264 and 267, change ‘in the end’ to ‘at the end’.

Line 294, ‘Ca’ and ‘Mg’ should be ‘Ca2+’ and ‘Mg2+’.

Line 299, change ‘the biofouling’ to ‘biofouling’.

Line 321, add ‘extracellular polymeric substances’ and remove ‘EPSs’

I strongly recommend that the authors use the Oxford comma throughout the manuscript.

  1. Please begin the Introduction section with a description of the working principles of an osmotic microbial fuel cell.

  1. Citation 15 is irrelevant to this manuscript. It talks about copper nanoparticles instead of silver nanoparticles. Please remove it and add a relevant article.

  1. In Paragraph 2 of the Introduction section, the authors should briefly discuss on how AgNPs mitigate biofouling. Please consider these recommended articles and cite them in the Introduction section as well as in Lines 321–325:

Polysulfone Membranes Modified with Bioinspired Polydopamine and Silver Nanoparticles Formed in Situ To Mitigate Biofouling. Environ. Sci. Technol. Lett. 2015, 2 (3), 59-65.

In situ immobilization of silver nanoparticles for improving permeability, antifouling and anti-bacterial properties of ultrafiltration membrane. J. Membr. Sci. 2016, 499, 269-281.

Preparation and characterization of antifouling and antibacterial polysulfone ultrafiltration membranes incorporated with a silver-polydopamine nanohybrid. J. Appl. Polym. Sci. 2018, 135 (27).

Hierarchically structured Janus membrane surfaces for enhanced membrane distillation performance, ACS Appl. Mater. Interfaces 2019, 11, 25524-25534.

  1. In Figure 1, I recommend that the authors include chemical structures to clearly illustrate how AgNPs were immobilized on the membrane through the various modification steps.

  1. Please group Sections 2.2 and 2.4 into one section.

  1. Please include scale bars in the SEM images (Figures 2 and 6). Please also include elemental EDX mapping images for both membranes in Figure 2 to confirm that the particle clusters were indeed AgNPs.

  1. Please recapture the CLSM image of the modified membrane in Figure S3. The quality of the existing image is unacceptable.

  1. I strongly recommend that the authors subject the modified membranes to ultrasonic treatment (or other treatments) to prove that the AgNPs are strongly immobilized on the membrane surface.

  1. Figure 3(c) is very confusing. Please use filled and empty symbols to represent voltage and power density, respectively.

  1. Please provide detailed explanation(s) on the mechanism(s) of AgNPs in bringing about better electricity generation. Do not merely report on the observations.

  1. What is the mass density of immobilized AgNPs before and after the experiment?

  1. In Line 290, the author mentioned that the fouling layer was thicker on the membranes used in the control OsMFC experiment. Please provide evidence on this (i.e., cross-sectional SEM images). The determination of thickness using CLSM did not seem very convincing.

  1. The alignment of sub-figures in Figure 6 seems a little off. Please correct that.

  1. In Figure S4, please indicate what each color represents.

  1. Why were the modified membranes less effective in mitigation the deposition of ß-D-glucopyranose polysaccharides?

Author Response

(The authors gave the same response as above.)

Round 2

Reviewer 1 Report

The authors replied extensively to all the coments with major revision of the document.